# Strength Enhancement of Interlocking Hollow Brick Masonry Walls with Low-Cost Mortar and Wire Mesh

Panuwat Joyklad [1], Nazam Ali [2], Muhammad Usman Rashid [2], Qudeer Hussain [3], Hassan M. Magbool [4], Amr Elnemr [5] and Krisada Chaiyasarn [6,*]

1 Department of Civil and Environmental Engineering, Faculty of Engineering, Srinakharinwirot University, Bangkok 26120, Thailand; panuwatj@g.swu.ac.th
2 Department of Civil Engineering, School of Engineering, University of Management and Technology, Lahore 54770, Pakistan; nazam.ali@umt.edu.pk (N.A.); usman.rashid@umt.edu.pk (M.U.R.)
3 Center of Excellence in Earthquake Engineering and Vibration, Department of Civil Engineering, Chulalongkorn University, Bangkok 45142, Thailand; ebbadat@hotmail.com
4 Civil Engineering Department, Faculty of Engineering, Jazan University, Jazan 45142, Saudi Arabia; h.magbool@jazanu.edu.sa
5 Civil Engineering Program, German University in Cairo, New Cairo City 11835, Egypt; amr.elnemr@guc.edu.eg
6 Thammasat Research Unit in Infrastructure Inspection and Monitoring, Repair and Strengthening (IIMRS), Thammasat School of Engineering, Faculty of Engineering, Thammasat University Rangsit, Klong Luang 12121, Thailand
* Correspondence: ckrisada@engr.tu.ac.th

**Abstract:** Cement–clay Interlocking Hollow Brick Masonry (CCIHBM) walls are characterized by poor mechanical properties of bricks and mortar. Their performance is observed to be unsatisfactory under both gravity and seismic loads. There is an urgent need to develop sustainable, environmentally friendly, and low-cost strengthening materials to alter the structural behaviour of brick masonry walls in terms of strength and ductility. The results of an experimental investigation conducted on the diagonal compressive response of CCIHBM walls are presented in this study. In this experimental study, a total of six CCIHBM walls were constructed using cement–clay interlocking hollow bricks. One was tested as a control or reference wall, whereas the remaining walls were strengthened using cement mortar. In some walls, the cement mortar was also combined with the wire mesh. The research parameters included the type of Ordinary Portland Cement (OPC) (Type 1 and Type 2), thickness of cement mortar (10 mm and 20 mm), and layers of wire mesh (one and three layers). The experimental results indicate that control or unstrengthened CCIHBM walls failed in a very brittle manner at a very low ultimate load and deformation. The control CCIHBM wall, i.e., W-CON, failed at an ultimate load of 247 kN, and corresponding deflection was 1.8 mm. The strength and ductility of cement mortar and wire mesh-strengthened walls were found to be higher than the reference CCIHBM wall. For example, the ultimate load and deformation of cement-mortar-strengthened wall were found to be 143% and 233% higher than the control wall, respectively. Additionally, the ultimate failure modes of cement mortar and wire mesh strengthened were observed as ductile as compared to the brittle failure of reference wall or unstrengthened CCIHBM wall, which increased by 66% and 150% as compared with the control wall.

**Keywords:** brick; cement; clay; strengthening; mortar; wire mesh

## 1. Introduction

Natural disasters such as earthquakes, landslides, liquefaction of ground, and tsunamis cause widespread destruction and damage to infrastructure such as public and commercial buildings, roads, and bridges. Among these natural disasters, the damage due to earthquakes is most common around the world. During earthquakes, the ground shaking may cause complete or partial damage to roads, bridges, and buildings. Complete damage

to buildings may result in the highest number of causalities compared to partial damage [1,2]. An appropriate selection of material and proper design is vital to safeguard the infrastructure against all types of natural disasters, especially earthquakes. Brick and/or block masonry structures (both un-reinforced and reinforced) are commonly constructed throughout the world because of the wide availability of the construction materials and comparatively low construction cost. In some countries, concrete blocks of different shapes are frequently used for construction. In Asian and Southeast Asian regions, clay brick masonry construction is very common. Different waste materials such as ceramics, fly ash, and slags have been frequently used in the past to produce bricks [3–5]. However, there are few drawbacks of masonry construction such as weak joints, brittle nature, and insufficient lateral stability, especially in the case of un-reinforced masonry construction. As a result, widespread destruction to the masonry structures was observed in past earthquakes. For example, an earthquake of 6.3 magnitude occurred in Christchurch, New Zealand, in 2011, which caused massive destruction. Although this earthquake was moderate, it caused devastating damage to the masonry structures and buildings because of the high shaking of building levels in the city centre. In the damaged buildings, those that were unreinforced masonry structures suffered the highest damage among other buildings in that earthquake [6]. Although Thailand is located in a low-seismic region in Southeast Asian and is far away from the sources that may case high intensity earthquakes, in the past, few devasting earthquakes have been recorded in the northern provinces of Thailand such as Chiang Mai and Chiang Rai. In 2004, a 9.1 magnitude earthquake was recorded near the Island of Sumatra, Indonesia. This earthquake resulted in a devasting tsunami that killed almost 225,000 people in several countries such as Sri Lanka, Maldives, and Thailand. After the earthquake, the water level was observed up to 19.6 m at Ban Thung Dap and 15.8 m at Ban Nam Kim, Thailand. Damage to the infrastructure was mainly observed in two districts, i.e., Chiang-rai and Chiang-saen, as shown in Figure 1 [7].

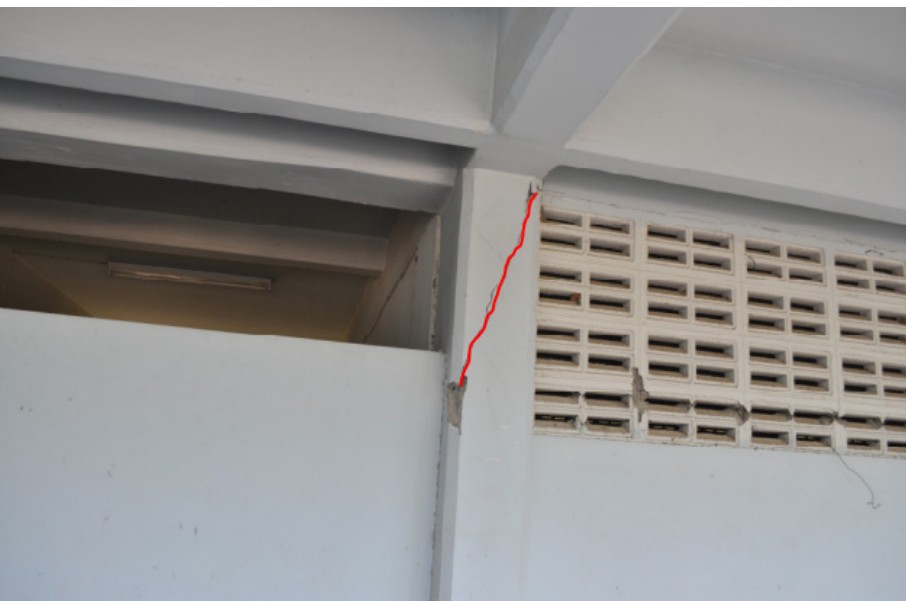

**Figure 1.** Damage of a building with small columns in Chiang-Saen District [7].

　　　Brick masonry is vital part of construction in Thailand, along with other countries in the world. Different types of bricks are used for residential, commercial, educational, and religious infrastructure [8–13]. The salient advantages of brick masonry construction are high load bearing capacity, the use of local materials, and superior resistance against extreme weather conditions [14]. Additionally, brick masonry construction is usually considered more serviceable, durable, and environmentally friendly [15,16]. However,

existing earthquakes have shown severe damage to masonry construction as shown in Figures 2 and 3.

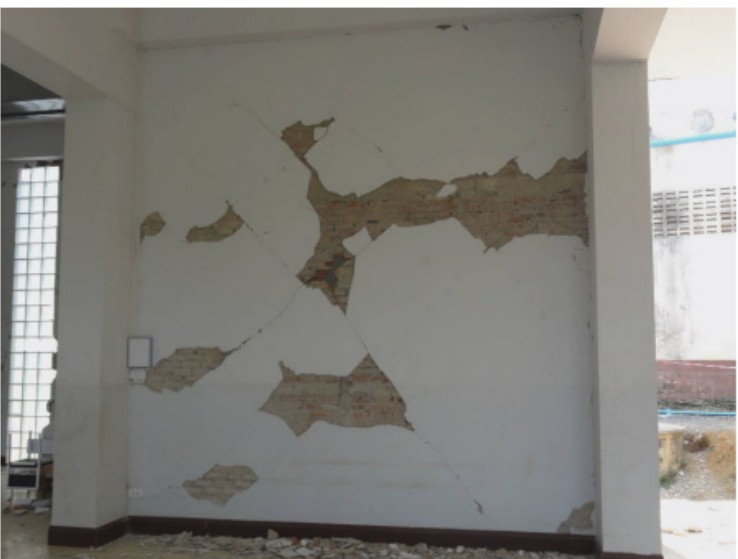

**Figure 2.** Diagonal cracks in a masonry wall [7].

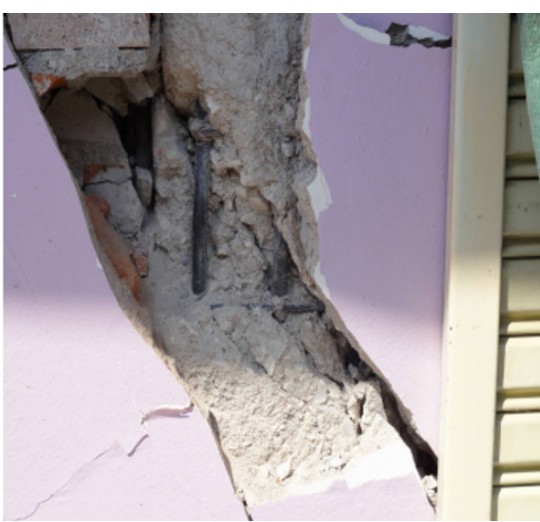

**Figure 3.** Damage due to load transfer from masonry walls [7].

Therefore, there is an urgent need to enhance the lateral stability and load carrying capacity of brick masonry construction. Advanced Fibber-Reinforced Polymer (FRP) composites and other materials have been extensively utilized for the structural repair and strengthening of masonry walls [17–23]. FRP composites are lightweight and high in tensile strength. Tumialan et al., 2001, investigated the use of different FRP systems to enhance the strength of un-reinforced masonry walls. In their study, a total number of six block masonry walls were constructed and tested up to ultimate failure. The test results indicate that the use of FRP systems is very useful to alter the structural performance of the block masonry walls [24]. In another study, Iii et al., 2001 used glass FRP composites to alter the flexural capacity of concrete block walls. The glass FRP composites were applied in the vertical direction, i.e., perpendicular to the bed joints. Based on experimental results, the authors reported two types of failures, i.e., glass FRP fracture and combination of fracture and de-bonding. The flexural capacity of glass-FRP-strengthened walls was found to be higher than the control walls [25]. Bui et al., 2015, studied the behaviour of FRP-strengthened hollow concrete brick masonry walls. The masonry walls were strengthened using glass

and textile FRP composites. A total of six walls were constructed and tested under lateral loading. The FRP composites were applied in different strengthening configurations. All types of FRP composites were found to be feasible to extend the structural integrity of the hollow concrete brick masonry walls. The performance of textile FRP composite was found to be lower than the glass FRP composite [26]. The use of FRP composites was also found effective to enhance the strength and ductility of solid and hollow clay brick masonry walls [27,28]. Hamoush et al., 2014, used carbon FRP composites to strengthen fired clay brick masonry walls. In their study, a total of fifteen walls were constructed and tested. Twelve walls were strengthened using carbon FRP composites and three were considered as control. The walls were tested under static load. The maximum load of CFRP strengthened wall was found to be 29.72 kPa, whereas the maximum load of control and/or unstrengthened masonry walls was found to be 1.43 kPa [29].

Although FRP composites are very beneficial to alter the load-carrying capacity of masonry walls, such FRPs are very expensive. There is a need is to explore the use of low-cost and locally available materials that can be used to alter the performance of brick walls. These materials are more efficient in terms of cost and strength. The total strengthening cost by FRP composites is usually 100–130% higher than the mortar or concrete jacketing [30]. In Thailand, the use of cement clay interlocking hollow bricks are very common for masonry structures due to their salient features such as light weight, durability, and cost-efficiency. Traditionally, CCIH bricks are staked over each other in such a way that interlocks are responsible for bond strength. Past studies have investigated mechanical properties of CCIH bricks [31,32]. Joyklad and Hussain (2018 and 2019) investigated the behaviour of CCIH brick masonry walls under diagonal and axial compression. The results indicate that the ultimate failure of CCIH brick masonry walls is very vulnerable, especially when CCIH brick masonry walls were construed in traditional manner [33–35]. Recently, Joyklad and Hussain, 2020, tested the performance of the CCIH brick masonry walls under earthquake loads. The results indicate that the lateral ductility of the CCIH brick masonry walls is very low [36]. A detailed review of the existing studies indicates that so far, no study has been conducted on the strengthening of the CCIH brick masonry walls by using low-cost and locally available materials such as cement mortar and wire mesh. Therefore, the current research work was mainly proposed to investigate the feasibility of the different traditional materials to alter the structure behaviour of the cement clay interlocking hollow brick masonry walls in terms of ultimate strength and deformation.

## 2. Details of Experimental Program

In this study, a total number of six walls were constructed and tested. The research parameters included were the type of Ordinary Portland Cement (OPC) (Type 1 and Type 2), thickness of cement mortar (10 mm and 20 mm), and layers of wire mesh (one and three layers). The OPC Type 1 cement was ordinary Portland cement of Type 1, whereas the OPC Type 2 cement was high-performance non-shrink cement. The names of CCIHBL wall specimens, and research parameters are given in Table 1. The masonry wall W-CON was constructed in a traditional way, and holes in cement clay interlocking bricks were filled with the cement mortar of Type 1. The construction method of the second wall (W-OPC1-10) was also similar to the control wall; however, cement mortar was also applied on both external faces. The thickness of external cement mortar was chosen to be 10 mm. In the masonry wall W-OPC1-20, the thickness of external cement mortar was increased to 20 mm. In the masonry walls W-OPC2-20, the type of external cement mortar was changed to Type 2 and external mortar thickness was 20 mm. In masonry walls W-OPC1-10-1W and W-OPC1-10-3W, wire mesh was also attached to external surface of masonry walls prior to the external cement mortar. For each parameter a single was constructed and tested under axial compression.

**Table 1.** Details of CCIHBM walls.

| Wall Specimen | Strengthening Material | Thickness (mm) | Type of Cement |
|---|---|---|---|
| W-CON | - | - | - |
| W-PC1-10 | Cement mortar | 10 | OPC Type 1 |
| W-PC1-20 | Cement mortar | 20 | OPC Type 1 |
| W-PC2-20 | Cement mortar | 20 | OPC Type 2 |
| W-PC1-10-1W | Cement mortar + Wire Mesh | 10 | OPC Type 1 |
| W-PC1-10-3W | Cement mortar + Wire Mesh | 20 | OPC Type 1 |

## 3. Dimensional Details of Masonry Walls

The size of CCIHBM walls was selected in such a way to accommodate the capacity and size of reaction frame in the laboratory. The length of CCIHBM walls 1000 mm and height were also kept equal to the length, as shown in Figure 4, and the thickness of the walls was equal to the thickness of cement clay interlocking bricks, i.e., 125 mm. Locally available CCIH bricks were used. A typical sample of cement clay interlocking brick is shown in Figure 5. The type of bond was considered as the running bond for the construction of the CCIHBM walls (Figure 6), and circular holes were filled with the cement mortar. This type of construction is very common in Thailand.

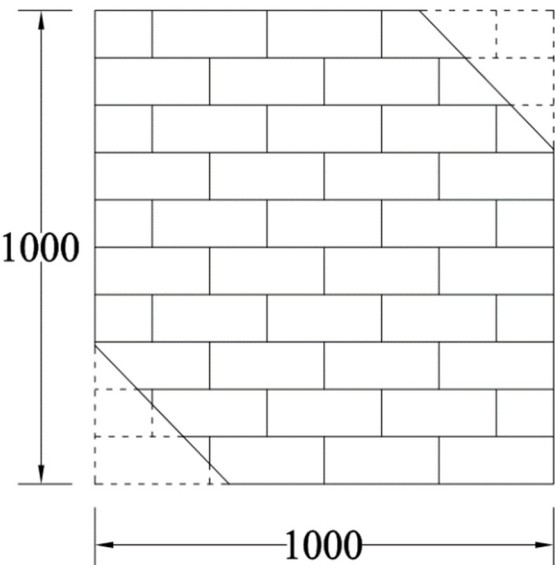

**Figure 4.** Dimensional details of wall specimens (units: mm).

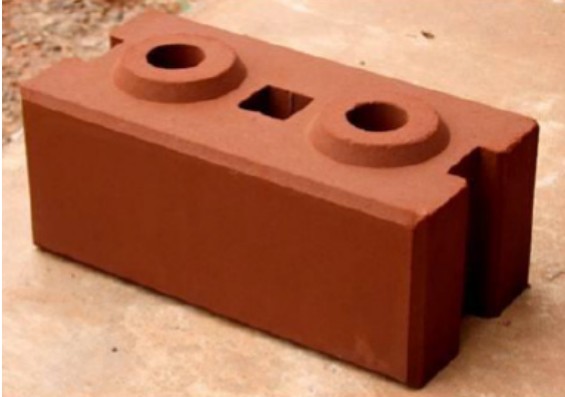

**Figure 5.** CCIH brick.

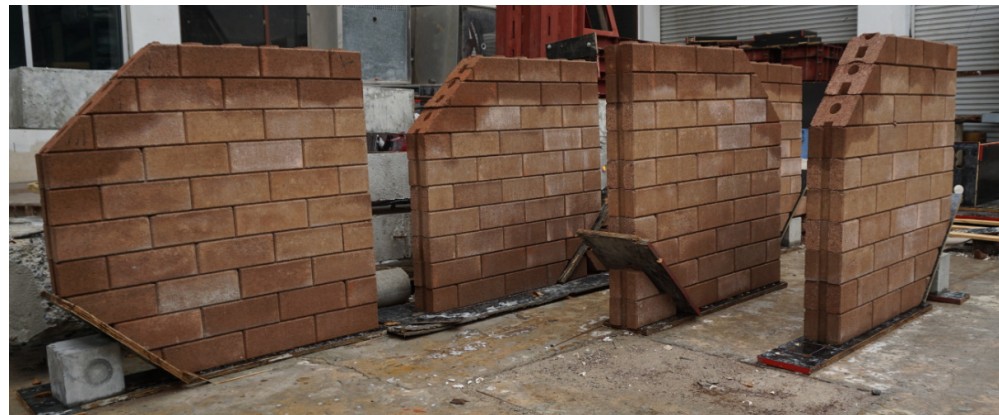

**Figure 6.** Typical construction of CCIHBM walls.

## 4. Materials

In this study, two types of ordinary Portland cements, i.e., OPC Type 1 and OPC Type 2, were used to prepare cement mortar. The OPC Type 1 cement was ordinary Portland cement of Type 1, whereas the OPC Type 2 cement was high-performance non-shrink cement. The compressive strength of cement mortar with OPC Type 1 was 25 MPa, and the compressive strength of cement mortar with OPC Type 2 was 50 MPa. For the construction of masonry walls, the CCIH bricks were obtained from the local manufacturers. Standard compression and water absorption tests were performed to obtain the mechanical properties of CCIH bricks. The average compressive strength was 6.74 MPa, and water absorption capacity was 8.80%.

## 5. Strengthening of CCIHBM Walls

The strength enhancement of CCIHBM walls was performed by using cement mortar with and without wire mesh to alter the structural behaviour of CCIHBM walls. The cement mortar was prepared using ordinary Portland cement of Type 1 and Type 2 (PC1 and PC2) and sand. The CEMENT mortar was mixed by using a ratio of 1:2 (cement:sand) to achieve CEMENT mortar of maximum compressive strength. The CEMENT mortar was applied to the CCIHBM walls by using a simple hand layup technique. The application of CEMENT mortar on CCIHBM wall is shown in Figure 7. In CCIHBM wall specimens W-PC1-10-1W and W-PC1-10-3W steel wire mesh (Figure 8) were also fixed to the bricks prior to the application of the CEMENT mortar. The installation of the wire mesh on the CCIHBM wall is shown in Figure 9, and a typical CEMENT mortar strengthened CCIHBM wall is shown in Figure 10.

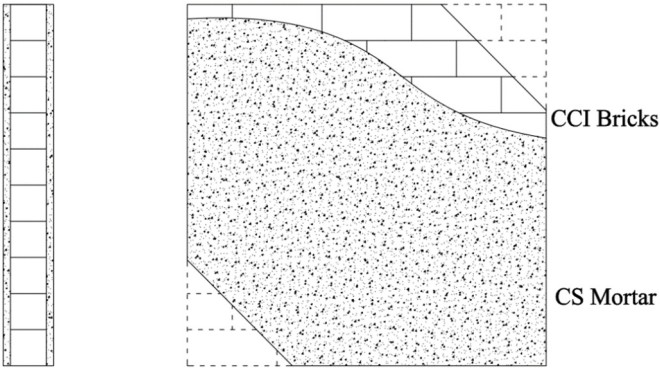

**Figure 7.** The application of CEMENT mortar on CCI bricks.

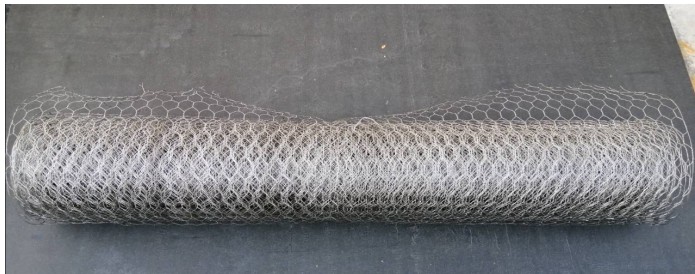

**Figure 8.** Typical view of wire mesh.

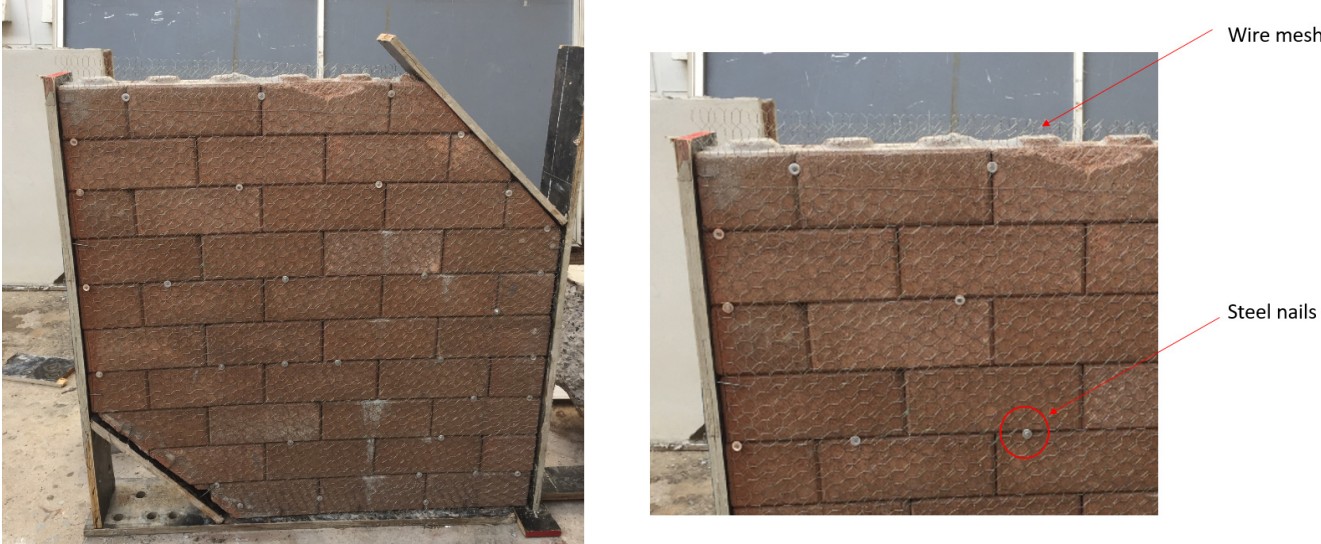

**Figure 9.** The attachment of wire mesh to the CCIH bricks.

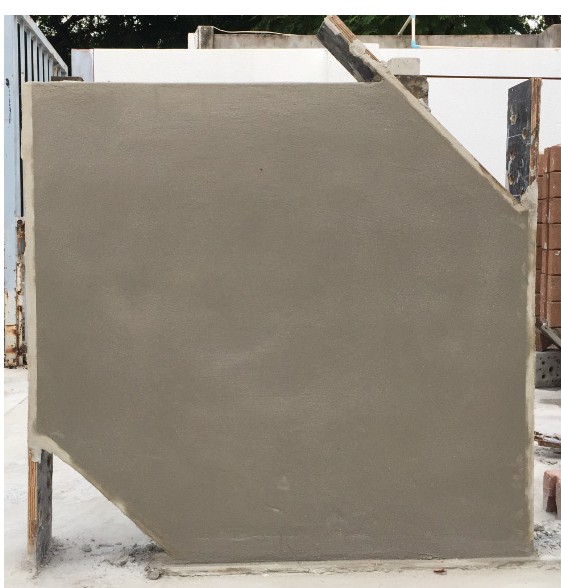

**Figure 10.** CCIH brick masonry wall with CEMENT mortar.

## 6. Load and Instrumentation

The CCIHBM walls were tested in diagonal compression. A rigid steel reaction frame was used to apply the load at the top of the CCIHBM walls. The maximum capacity of reaction frame was 2000 kN. The load was applied using a hydraulic jack as shown in

Figure 11. A pre-calibrated load cell of capacity 1000 kN was placed under the piston of the hydraulic jack to record the load. A steel plate 20 mm in thickness was placed at the top of the CCIHBM wall to safeguard the uniform application of the load. A total of four linear variable differential transducers (LVDT) were installed at different locations to measure the deformation of the CCIHBM walls under the applied load. The details of loading setup and installation of LVDTs are shown in Figures 11 and 12. Additionally, during the test, the appearance and propagation of the cracks was carefully marked and captured by using digital cameras.

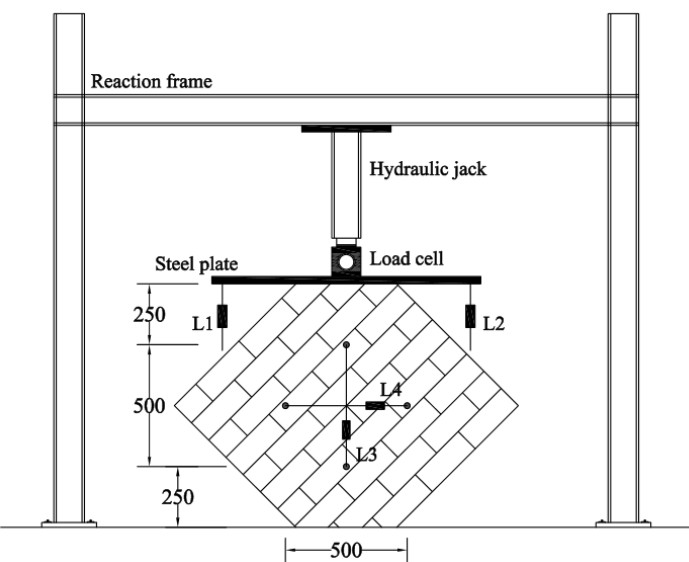

**Figure 11.** Loading setup and instrumentation details (units in mm).

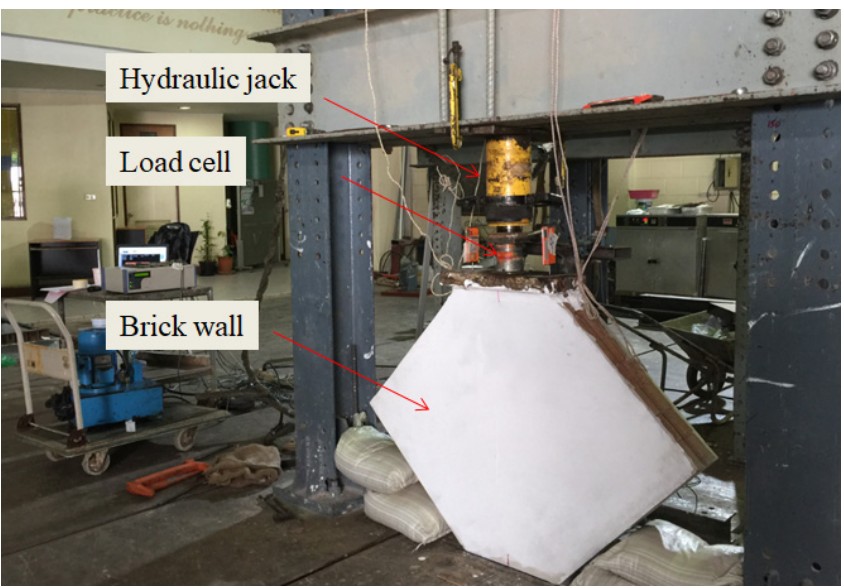

**Figure 12.** Loading setup.

## 7. Results and Discussion

### 7.1. Axial Load versus Deformation Responses

In this study, a total of six CCIHBM walls were constructed and tested under diagonal compression loading. Out of the six, five walls were externally strengthened using combination of cement mortar and wire mesh, whereas one CCIHBM wall was tested without any strengthening material to serve as control wall. The experimental results are given

in Table 2. The experimental results in terms of diagonal load versus deformation are shown in Figures 13–16. From Figures 13–16, it can be noted that the strength and ductility of cement mortar and wire-mesh-strengthened walls were found to be higher than the reference CCIHBM wall. A linear response (load versus deflection) was noticed for the control wall specimen, i.e., W-CON. The crushing of the CCIH bricks was observed at the ultimate load following a sudden drop in the load. The initial stiffness of cement mortar and wire mesh strengthened walls were also found to be higher than the reference wall as shown in Table 2.

**Table 2.** Summarized test results.

| Test Walls | Load (kN) | Deflection (mm) | Initial Stiffness (kN/mm) |
|---|---|---|---|
| W-CON | 247 | 1.8 | 235 |
| W-PC1-10 | 278 | 4.0 | 295 |
| W-PC1-20 | 336 | 2.0 | 414 |
| W-PC2-20 | 510 | 2.2 | 500 |
| W-PC1-10-1W | 410 | 4.5 | 498 |
| W-PC1-10-3W | 600 | 6.0 | 524 |

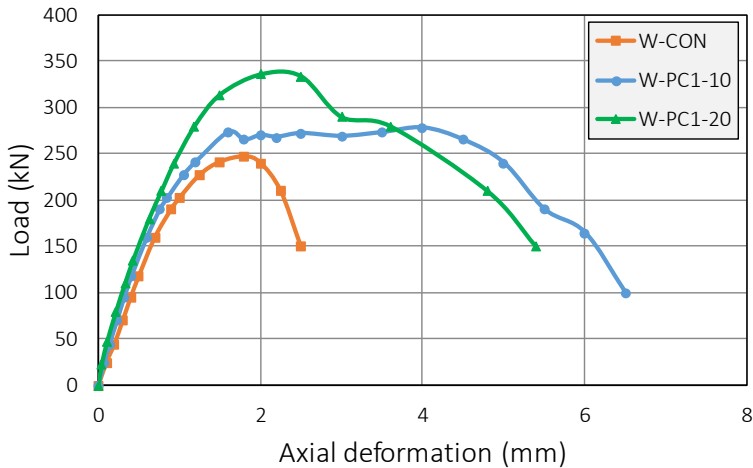

**Figure 13.** Experimental responses (Control and PC1 walls).

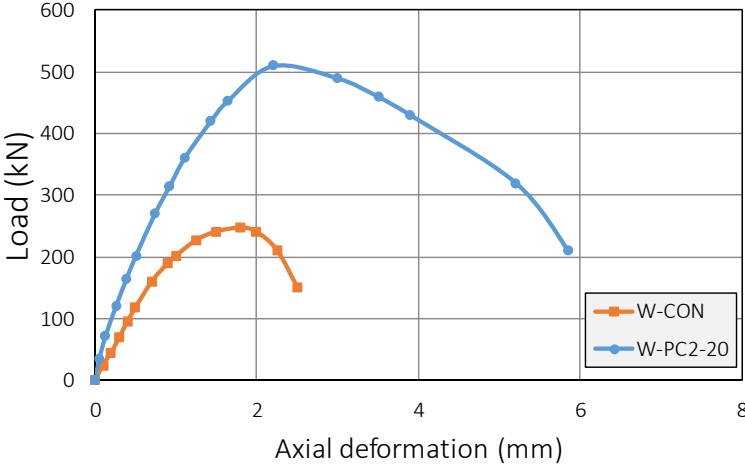

**Figure 14.** Experimental responses (Control and PC2 walls without wire mesh).

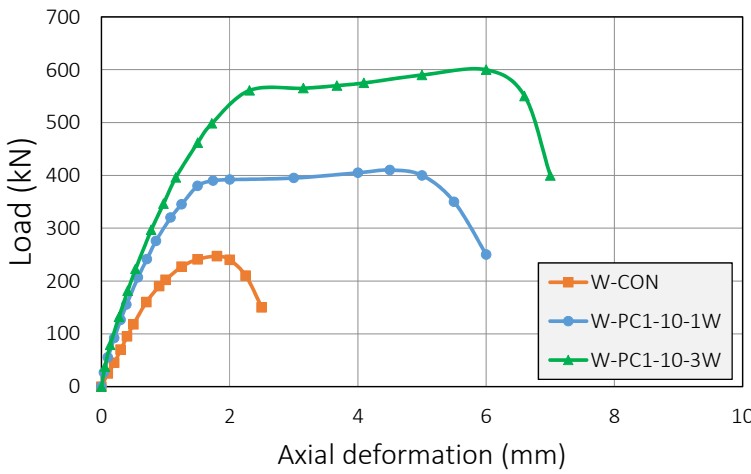

**Figure 15.** Experimental responses (Control and PC1 walls with wire mesh).

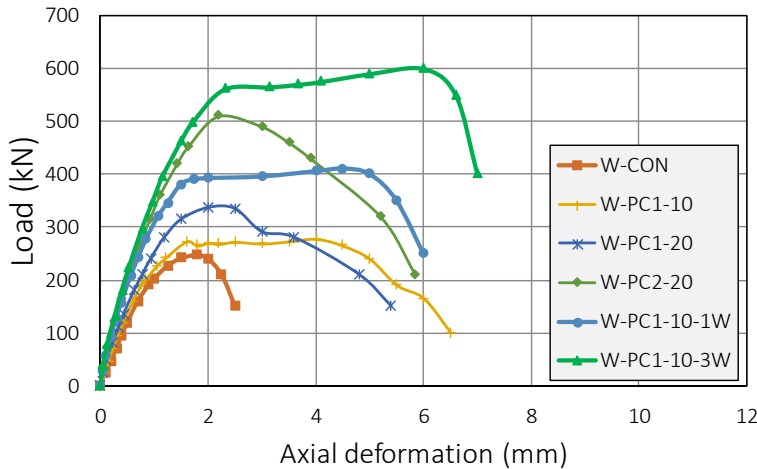

**Figure 16.** Experimental responses of all CCIHBM walls.

On the other hand, a gradual drop in the load carrying capacity was observed after the peak strength in CEMENT-mortar-strengthened CCIHBM walls such as W-PC1-10, W-PC1-20, and W-PC2-20. The use of wire mesh is found to be very effective to alter the diagonal load versus deformation responses of CCIHBM walls. In all cement-mortar-strengthened walls, W-PC1-10, W-PC1-20, and W-PC2-20 (without wire mesh), the load-versus-deformation responses were mainly linear up to the first peak, except the wall specimen W-PC1-10. However, in the case of CCIHBM walls W-PC1-10-1W and W-PC1-10-3W (strengthened using cement mortar and wire mesh), the axial-load-versus-axial-deformation responses were tri-linear. In the tri-linear responses, the first part is identical to the load-versus-deformation response of the unstrengthened wall, i.e., W-CON. The second part represents the transition curve, and the third part is once again linear. However, the stiffness of the third part is much lower than that of the first part. This tri-linear behavior is an indication that the use of wire mesh is very suitable to provide the external confinement to the CCIHBM walls.

### 7.2. Load Behavior of CCIHBM Walls

The load behaviors of the CCIHBM walls are shown Figure 17. The control CCIHBM wall, i.e., W-CON, failed at an ultimate load of 247 kN. The ultimate load-carrying capacity of the CEMENT mortar (Portland cement Type 1)-strengthened walls W-PC1-10 and W-PC1-20 was enhanced by 13% and 36% compared to the control wall, i.e., W-CON. The ultimate load carrying capacity of cement (Portland cement Type 2) was recorded higher than the walls W-PC1-10 and W-PC1-20. This phenomenon could be related to the higher

compressive strength of the Portland cement of Type 2 as compared to the compressive strength of Portland cement Type 1. The ultimate load of wall W-PC2-20 was recorded to be 106% higher than the control wall (W-CON). Further, it was found that the use of wire mesh is also very effective to delay the cracking of the CCIHBM walls and to further enhance the ultimate load-carrying capacity of CCIHBM walls. In the case of CCIHBM walls W-PC1-10-1W and W-PC1-10-3W (strengthened with cement mortar and wire mesh), the ultimate load-carrying capacity was increased by 66% and 143%, respectively, as compared to the control wall, i.e., W-CON.

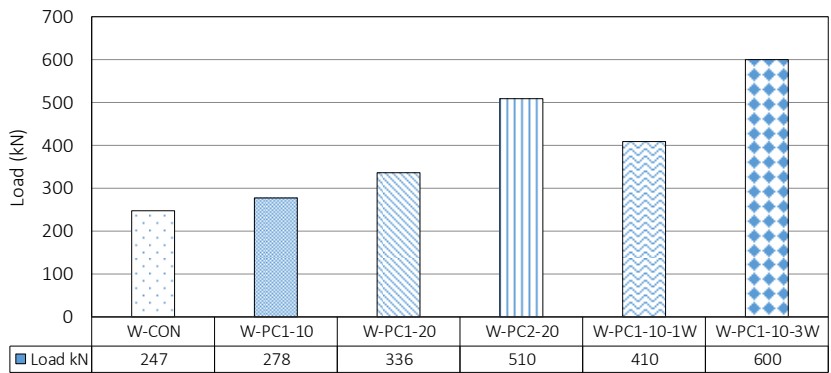

**Figure 17.** Load of CCIHBM walls.

### 7.3. Ductility of the CCIHBM Walls

The experimental results in terms of ultimate deformation again peak diagonal load are graphically shown in Figure 18. It can be seen that the use of cement mortar with and without wire mesh is very effective to enhance the ultimate deformation or ductility of the CCIHBM walls. The control wall, i.e., W-CON, failed at an ultimate axial deformation of 1.80 mm. The ultimate axial deformation of cement (Portland cement Type 1)-mortar strengthened walls W-PC1-10 and W-PC1-20 was enhanced by 122% and 11%, respectively, compared to the control wall. The ultimate axial deformation of cement (Portland cement Type 2) was recorded as higher than the walls W-PC1-10 and W-PC1-20. This phenomenon could be related to the higher compressive strength of the Portland cement Type 2 as compared to the compressive strength of Portland cement Type 1. The ultimate axial deformation of wall W-PC2-20 was recorded as 22% higher than the control wall. Further, it was found that the use of wire mesh is also very effective to further enhance the ultimate axial deformation of CCI walls. In the case of CCIHBM walls strengthened with cement mortar and wire mesh, i.e., W-PC1-10-1W and W-PC1-10-3W, the ultimate axial deformation was increased by 150% and 233% compared to the control wall, i.e., W-CON.

### 7.4. Failures of CCIHBM Walls

The failures of CCIHBM walls are shown in Figures 19–22. In the case of the control wall, i.e., W-CON, the ultimate failure was mainly due to the splitting of the bricks at the middle of the CCIHBM wall, as shown in Figure 19. Prior to the ultimate failure of the control wall, slight splitting and cracking of cement clay interlocking bricks was observed at the bottom of CCIHBM wall. At that moment, severe crushing of the bricks was also observed at the bottom edge of CCIHBM walls. The ultimate failure modes of cement-mortar-strengthened walls (without wire mesh) were approximately similar to that of the control wall; however, compression crushing of the bricks was not observed due to the presence of cement mortar. In these walls, the peeling of cement mortar was observed at the bottom of the CCIHBM wall, as shown in Figure 20. In contrast to the walls strengthened with cement mortar without wire mesh, the ultimate failure modes of CCIHBM walls (strengthened using cement mortar with wire mesh) were less explosive and more ductile as shown in Figure 21. In these walls, fracture of wire mesh and peeling of cement mortar

were observed at the middle of the CCIHBM walls as shown in Figure 22. The ultimate failure modes of CCI-IBM walls are inconsistent with the previous studies [29,37].

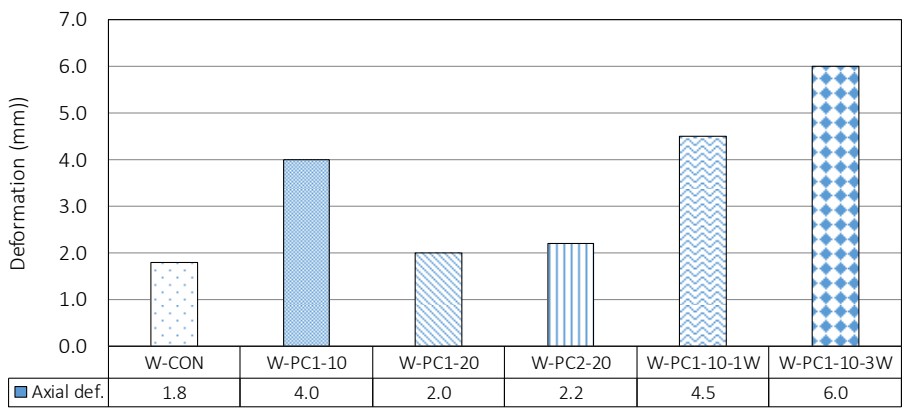

**Figure 18.** Deformation of CCIHBM walls.

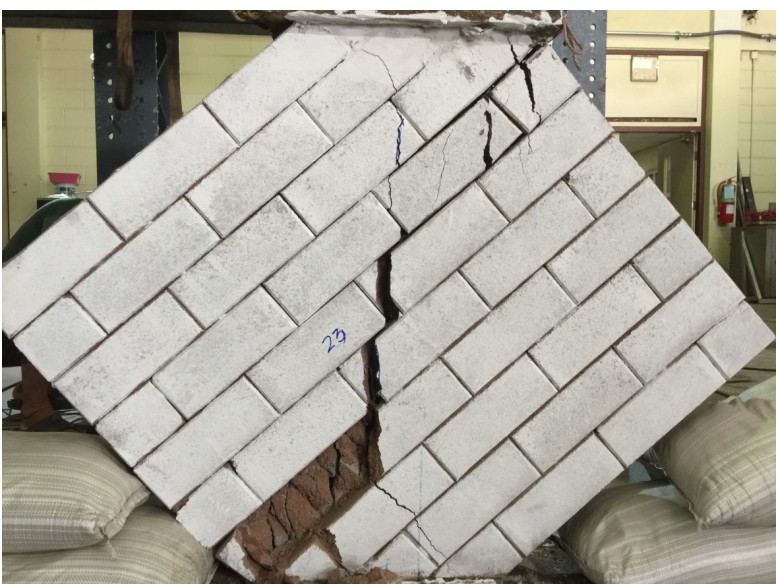

**Figure 19.** Ultimate failure mode of wall (W-CON).

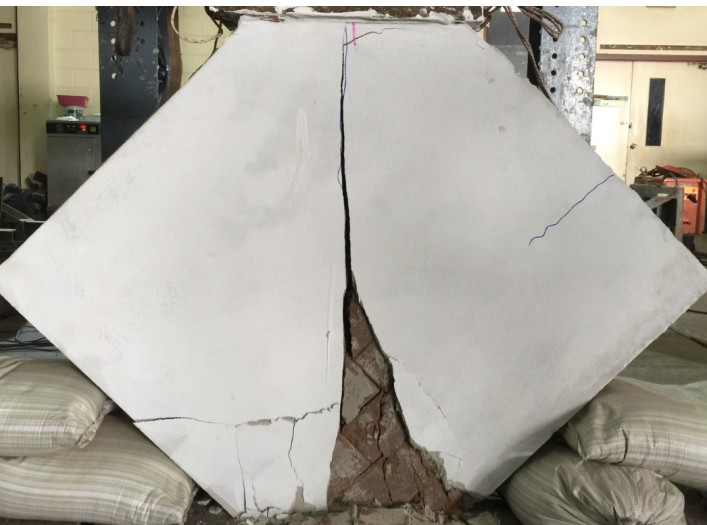

**Figure 20.** Failure of wall (W-PC1-10).

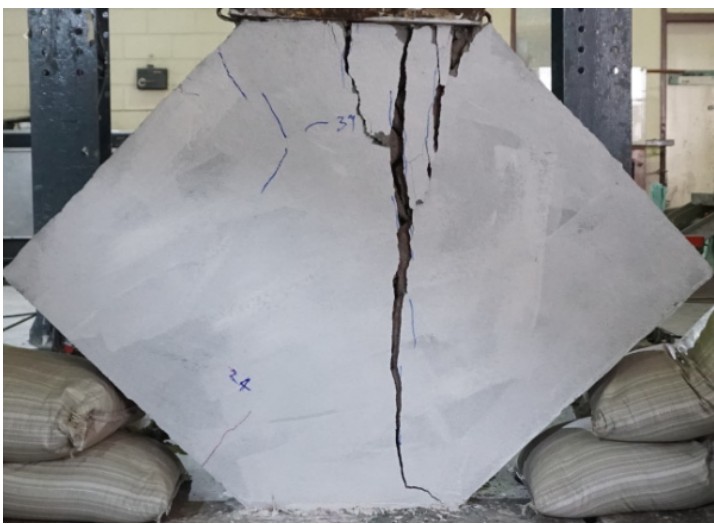

**Figure 21.** Failure of wall (W-PC2-20).

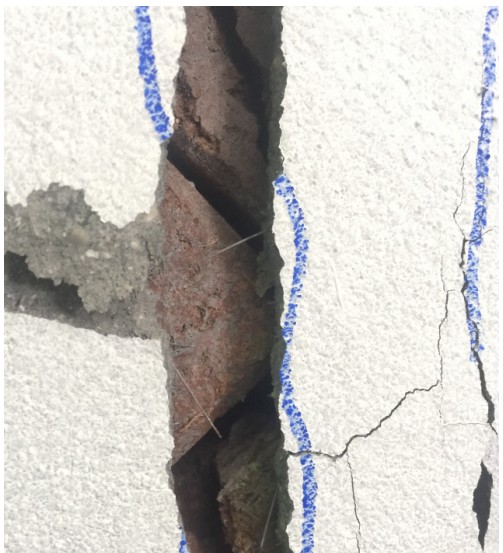

**Figure 22.** Fracture of wire mesh.

## 8. Conclusions

Based on experimental results, the following conclusions are derived:

1. The ultimate failure of control masonry wall was very brittle and sudden. The control CCIHBM wall, i.e., W-CON, failed at an ultimate load of 247 kN, and the corresponding deflection was 1.8 mm.
2. The ultimate failure modes of the CEMENT mortar with wire mesh strengthened CCIHBM walls were found to be ductile.
3. For the cement mortar and wire-mesh-strengthened walls, i.e., W-PC1-10-1W and W-PC1-10-3W, the ultimate axial deformation was increased by 150% and 233%, respectively, as compared to the control wall, i.e., W-CON.
4. The ultimate load carrying capacity of CCIHBM walls W-PC1-10-1W and W-PC1-10-3W was increased by 66% and 143%, respectively, as compared to the control wall, i.e., W-CON.
5. Based on experimental results, it can be concluded that the use of CEMENT mortar and wire-mesh is practical. However, there is need to evaluate and compare the performance of this method with other techniques.

6. Future studies also required to develop constitutive material models for CCIHBM walls strengthened with cement mortar and wire mesh using finite element analysis and analytical studies.

**Author Contributions:** Conceptualization, P.J., N.A., M.U.R., Q.H., H.M.M., A.E., and K.C.; Project administration, Q.H.; Writing—original draft, P.J., N.A., M.U.R., Q.H., H.M.M., A.E., and K.C.; Writing—review & editing, P.J., N.A., M.U.R., Q.H., H.M.M., A.E., and K.C. All authors have read and agreed to the published version of the manuscript.

**Funding:** The authors of this research work are very grateful to the Srinakharinwirot University, Thailand, for providing research grant (Research Grant ID 102/2563) to carry out the research work.

**Informed Consent Statement:** Not applicable.

**Data Availability Statement:** Not applicable.

**Acknowledgments:** The authors of this research work are very grateful to the Srinakharinwirot University, Thailand, for providing research grant (Research Grant ID 102/2563) to carry out the research work. Thanks are also extended to Asian Institute of Technology (AIT) for supporting test facilities.

**Conflicts of Interest:** The authors declare no conflict of interest.

## List of Abbreviations

| | |
|---|---|
| CCIHBM | Clay Interlocking Hollow Brick Masonry |
| CCIHB | Clay Interlocking Hollow Brick |
| OPC | Ordinary Portland Cement |
| FRP | Fibre Reinforced Polymers |

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
