# Peer review of "Strength Enhancement of Interlocking Hollow Brick Masonry Walls with Low-Cost Mortar and Wire Mesh"

_infrastructures, doi:10.3390/infrastructures6120166_

Round 1

Reviewer 1 Report

Dear Authors,

I have inserted my remarks into your text.

A reviewer

Reviewer 2 Report

  • The article needs extensive grammatical and syntax improvements. Use of English service center is required.
  • The abstract is written qualitatively. Majority of the qualitative statements should be modified for quantified result comparisons.
  • The authors mentioned about “An appropriate selection of material and proper design is very vital to safeguard the infrastructure against all types of natural disasters especially earthquake”. Following more recent works are recommended to be considered for this statement and added:
    • Experimental investigation of sound transmission loss in concrete containing recycled rubber crumbs.
    • Compressive behavior of concrete under environmental effects. IntechOpen.
    • .Temperature and humidity effects on behavior of grouts. Advances in concrete construction, 5(6), 659.
    • Nano silica and metakaolin effects on the behavior of concrete containing rubber crumbs. CivilEng, 1(3), 264-274.
  • The introduction needs to be revised for higher quality language. The authors mentioned some works without stating about the contributions, pros and cons and the how the current work would address. In addition, various waste material use in concrete should be refereed and mentioned briefly.
    In addition for the following statement “Advanced Fibber Reinforced Polymer (FRP) composites have been extensively utilized for the structural repair and strengthening of masonry walls’, the following reference should be considered:
    • Investigation of steel fiber effects on concrete abrasion resistance, Advances in concrete construction, 9(4), 367-374
  • The cost of use of FRP compared to simple procedure using no FRP should be elaborated.
  • What does it mean “The carbon FRPs were found effective to alter the load carrying capacity of masonry wall”
  • Figure should be smaller and if they don’t have much information, they should be removed.
  • Quantitative comparisons should be used instead of qualitative comparison :
    • The ultimate failure of control masonry wall was very brittle and sudden
  • Is there any information on the initial stiffness? The initial stiffness could be evaluated and included in tables.
  • Practical suggestion based on the financial and structural evaluation should be added.
  •  

Round 2

Reviewer 1 Report

Dear Authors,

I read your responses and revised manuscript. It is better but I am not satisfied totally.

I suggested the chapter 'Materials' whic was not created.

I still do not find characteristics of the fibers.

Abbreviation 'CS' mortar as cement sand mortar looks strange. According to obvious definition, each mortar consists of binder (in the this case it is cement), water and sand. So, there is no need to specify 'S' for sand.

I suggested to remove first sentences from the Conclusions giving the reason for removal but they still exist.

Conclusions have been improved but I still claim they are verbose.

Despite your statement that concrete is not so popular construction material in some Asian countries (?) you should write about popularity of that material in other countries. According to common knowledge, concrete due to cement as the basic constituent brings carbon dioxide emission. So, such idea has just come to my mind: why not to write about ceramics as more friendly to environment material? You mention cement but for mortars only. So, amount of CO2 emitted in such case is less. At least a reader has right to come to such conclusion. So, you may 'promote' ceramics although in many cases concrete is a definite 'winner' despite worse influence to environment.

Reviewer 2 Report

English language should be improved. 
